CellPress

## Commentary

# CanDIG: Federated network across Canada for multi-omic and health data discovery and analysis

L. Jonathan Dursi,[1,*] Zoltan Bozoky,[2] Richard de Borja,[3,13] Haoyuan Li,[4] David Bujold,[5] Adam Lipski,[4,6] Shaikh Farhan Rashid,[1] Amanjeev Sethi,[1] Neelam Memon,[4,24] Dashaylan Naidoo,[4,25] Felipe Coral-Sasso,[4,26] Matthew Wong,[7] P-O Quirion,[5] Zhibin Lu,[8] Samarth Agarwal,[9] Yuriy Pavlov,[1] Andrew Ponomarev,[4,10] Mia Husic,[11] Krista Pace,[1] Samantha Palmer,[1] Stephanie A. Grover,[12] Sevan Hakgor,[13] Lillian L. Siu,[13] David Malkin,[14] Carl Virtanen,[8] Trevor J. Pugh,[3,13,15] Pierre-Étienne Jacques,[16] Yann Joly,[17,18] Steven J.M. Jones,[4,19] Guillaume Bourque,[18,20,21] and Michael Brudno[1,22,23,*]

[1]DATA Team, University Health Network, Toronto, ON, M5G 2C4, Canada
[2]Providence Health Care, Vancouver, BC, V6Z 1Y6, Canada
[3]Ontario Institute of Cancer Research, Toronto, ON, M5G 0A3, Canada
[4]Canada's Michael Smith Genome Sciences Centre, BC Cancer Research Institute, Provincial Health Services Authority, Vancouver, BC, V5Z 4S6, Canada
[5]Canadian Centre for Computational Genomics, Montréal, QC, H3A 0G1, Canada
[6]Zymeworks, Vancouver, BC, V6H 3V9, Canada
[7]University of Waterloo, Waterloo, ON, N2L 3G1, Canada
[8]University Health Network, Toronto, ON, M5G 2C4, Canada
[9]University of Toronto, Toronto, ON, M5T 3A1, Canada
[10]GenX, Nottingham, NG1 1GF, UK
[11]Centre for Computational Medicine, The Hospital for Sick Children, Toronto, ON, M5G 1X8, Canada
[12]Genetics and Genome Biology Program, The Hospital for Sick Children, University of Toronto, Toronto, ON, M5G 1X8, Canada
[13]Princess Margaret Cancer Centre, University Health Network, Toronto, ON, M5G 2C1, Canada
[14]Division of Haematology/Oncology, The Hospital for Sick Children, Department of Pediatrics, University of Toronto, Toronto, ON, M5G 1X8, Canada
[15]Department of Medical Biophysics, University of Toronto, Toronto, ON, M5G 1L7, Canada
[16]Département de biologie, Université de Sherbrooke, Sherbrooke, QC, J1K 2R1, Canada
[17]Centre of Genomics and Policy, Department of Human Genetics, McGill University, Montreal, QC, H3A 0C7, Canada
[18]Department of Human Genetics, McGill University, Montreal, QC, H3A 0C7, Canada
[19]Department of Medical Genetics, University of British Columbia, BC, V6H 3N1, Canada
[20]Canadian Center for Computational Genomics (C3G), McGill University, Montreal, QC, H3A 0G1, Canada
[21]McGill Genome Center, Faculty of Medicine, McGill University, Montreal, QC, H3A 0G1, Canada
[22]Department of Computer Science, University of Toronto, Toronto, ON, M5T 3A1, Canada
[23]Vector Institute, Toronto, ON, M5G 1M1, Canada
[24]Present address: Vancouver, BC, Canada
[25]Present address: Molecular Genomics Laboratory, Providence Health & Services, Portland, OR, USA
[26]Present address: Florianopolis, 88037-400, Brazil
*Correspondence: jonathan.dursi@uhn.ca (L.J.D.), brudno@cs.toronto.edu (M.B.)

**We present the Canadian Distributed Infrastructure for Genomics (CanDIG) platform, which enables federated querying and analysis of human genomics and linked biomedical data. CanDIG leverages the standards and frameworks of the Global Alliance for Genomics and Health (GA4GH) and currently hosts data for five pan-Canadian projects. We describe CanDIG's key design decisions and features as a guide for other federated data systems.**

Canada is a confederation of provinces, each with its own health data privacy legislation, and data generated in each province must follow corresponding provincial laws. When we considered how to design a data sharing infrastructure for pan-Canadian human biomedical research projects, the diversity of regulations and legal frameworks across provinces meant there were very specific technical and privacy requirements, including (1) connecting distributed data under local control; (2) supporting data remaining on-premises; (3) simultaneously supporting multiple research in different domains, such as rare-disease and cancer research; (4) making use of existing compute, data, and authentication infrastructure as much as possible; (5) focusing first on enabling data discovery and querying, then analysis; and (6) a transparent open-source and standards-based approach for trust and interoperability.

While a common approach to data sharing is aggregation of large datasets in central repositories,[1] a federated approach was better suited to our framework across Canadian provinces.[2] Our requirements for transparent, open-source and standards-based approaches led us to adopting the international GA4GH technical and policy standards.[3] Implementing GA4GH standards in responsible data sharing (web resources), data security (web resources),

variant representation,[4] authentication,[5] and consents,[6] allowed our small team to quickly set up CanDIG, as well as to rapidly iterate on the platform and collaborate with groups internationally performing similar work and sharing lessons learned.

CanDIG is a Canadian national health research data platform, designed to support consented health research data discovery, querying, and analysis across centers and projects. CanDIG is the first multi-project human genomics and biomedical data federation in Canada, connecting the country's largest human sequencing centers (CGen; web resources). Deployed as a software stack at each site that joins the close governance of the federation, CanDIG has, to date, incorporated genomic and phenotypic data from five leading Canadian projects, including three projects spanning provincial boundaries: the Terry Fox Comprehensive Cancer Care Centre Consortium Network (TF4CN) and Terry Fox PRecision Oncology For Young peopLE (PROFYLE), the Canadian COVID-19 Genomics Network human sequencing project (CanCOGeN HostSeq), and two regional projects, POG (Personalized Onco-Genomics) and INSPIRE.[7] CanDIG participants include McGill University, The Hospital for Sick Children, University Health Network, Ontario Institute for Cancer Research, Canada's Michael Smith Genome Sciences Centre, Jewish General Hospital, and Université de Sherbrooke.

Here, we describe the choices we made building CanDIG—the data platform (the software stack and its operations) and the CanDIG Federation (the stakeholders, and the governance and policy between them that permits data access)—and compare them to other data federations for health research data. We first place CanDIG's platform in a three-dimensional landscape of data federations, considering range of queries, range of data types, and degree of decentralization, and compare it to well-known federated platform models such as the DataSHIELD,[8] Matchmaker Exchange,[9] Beacon network,[10] and the planned Federated EGA.[11] We then go into greater detail on the reason for our implementation choices and discuss technical details. Next, we describe the division of responsibilities and accountabilities in the federation, which is closely intertwined with the technical implementation.

In "CanDIG's implementation of GA4GH standards and technologies," we discuss the choice to adopt GA4GH standards, how those standards and collaborations allowed us to move faster and learn from other federations, and which standards we adopted immediately and what we have plans to adopt in the next version. We then discuss what a user has access to on the project dashboard and with the application programming interfaces (APIs) and conclude with future plans for developing and expanding CanDIG.

### Federated data platform models

Federated data systems span a variety of arrangements.[2] Here, we refer to federation in terms of the connection of "horizontal partitions" of data, connecting geographically separated research cohorts where the data for various participants can be found at multiple sites. We do not consider linking multiple separate data sources or types for the same data subject—clinical data in one store, genomic data in a second store, crossing "vertical partitions." In our model this happens internally to a site, and we refer to those operations as performing data integration, rather than federation. We also distinguish between data that is merely distributed, falling upon a user to discover, query, and assemble results by themselves, and data within a federated platform, where the nodes coordinate and communicate among each other.

One of the key parameters for a data federation is the degree of decentralization (Figures 1A–1C), which describes how queries flow through the system and whether there are centralized or distributed identities.

Federated data platform models can be considered along two additional dimensions: (1) the level of access they provide to the data and (2) the diversity of datasets accessible via the federation. Figure 1D illustrated the flexibility of data federations to handle additional constraints, such as adding differential privacy to a query. Figures 1E and 1F categorize several well-known health data federations along these dimensions. For example, the Datashield[8] and the Local EGA projects[11] are "central access" or "hub and spokes" models (Figure 1B) with a central infrastructure and identities. Data access can also be approached in

various ways, from having a few predefined queries, such as with the Beacon Network[10] and the Matchmaker Exchange (MME),[9] to running arbitrary analyses, such as with Datashield.[8]

Federated platforms must also be designed around the data types supported. Including a broader range of multi-omics, imaging, phenotypic and clinical data types is more valuable to researchers but increases complexity, may include more sensitive data, and makes federation governance a larger task.

### CanDIG federated platform model

Given our requirements, and learning from successful health data federations described above, we chose to implement a fully distributed federated data platform. The CanDIG platform has no centralized infrastructure or data; coordination occurs through the collaboration of the sites and the governance, policy, and standards decisions at the national level. This avoids a number of governance issues—such as the location and jurisdictional policies of centralized infrastructure—and makes it easier to assure local data custodians of full control over their data. An additional requirement was to support a range of querying and processing methods on a wide variety of data types. Figures 1E and 1F illustrate CanDIG's position in our federated data platform design space.

A more detailed look at the implementation of these other federated data platforms (see Table S2) demonstrated that to be consistent with our decentralized approach and our requirement to make use of existing infrastructure wherever possible, authentication would rely on the identities and authentication mechanisms of the participating sites. Users would log in with their home sites credentials rather than with a centralized CanDIG identity. Authorization decisions would have to be made locally at each site, based on the trusted federation-peer user identity and the nature of the request. Our requirement to allow a number of query and analysis methods over a plethora of different data types necessitate rather fine-grained authorization—allowing a user to access counts of data without necessarily allowing access to individual records, for instance, or allowing access to somatic cancer mutations

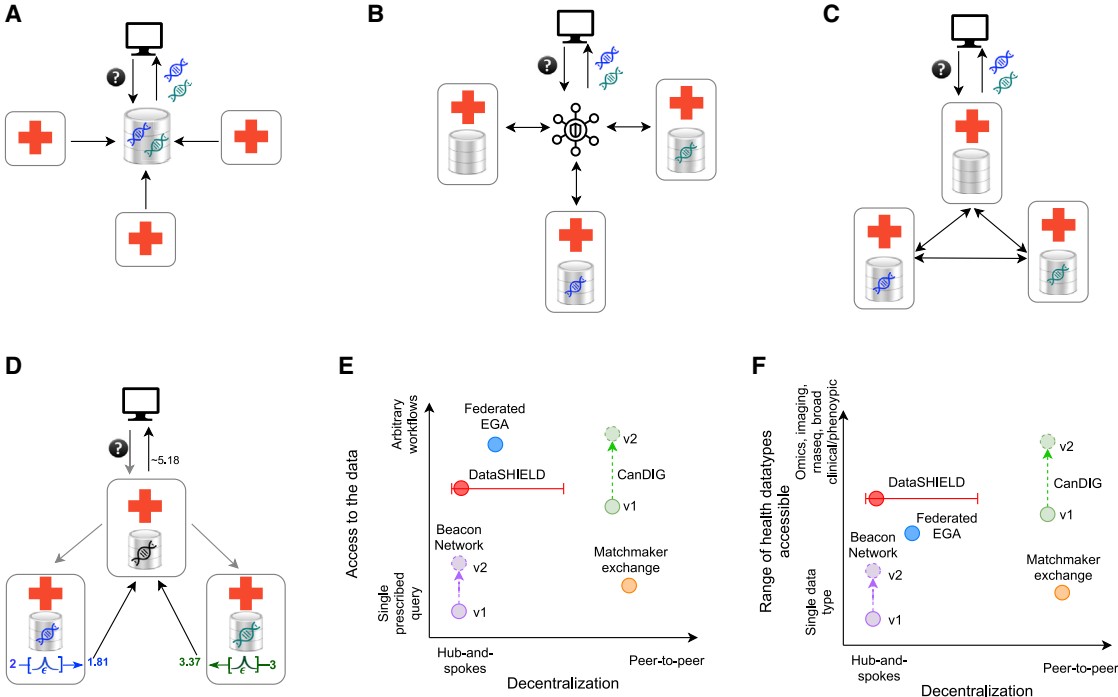

**Figure 1. Degree of decentralization of a federated data platform**

(A) A centralized (not federated) data repository where data is pulled manually or automatically into a central data store.

(B) A hub-and-spokes model of federation (a "central access model" as described in Thorogood et al.[2]), where there is significant central infrastructure that the peers are required to interact with.

(C) A decentralized, peer-to-peer network, where a user sends a request to a peer node—where relevant data may or may not be—and other peers are queried. Results (represented by DNA) are then returned to the user.

(D) Results can be locally processed at each site before being returned; in this case, perturbing results for the purposes of privacy enhanced analytics, in this case local differential privacy.[12] We can further categorize well-known health data federations along dimensions of decentralization (central-access to peer-to-peer).

(E) Level of access to data (pre-specified queries to arbitrary workflows).

(F) The range of different kinds of datasets supported by the various platforms.

but not germline variants. Details of our implementation can be found in supplementary notes section 1.

These requirements and design choices make for an approach quite different to other federated health data platforms. The requirement to support on-premises data meant moving to a centralized secure enclave in the cloud such as AnVIL[1] was not feasible. Unlike the Beacon network or DataShield, there is no central portal or infrastructure; unlike early versions of Local EGA, there is no central identity; unlike Matchmaker Exchange, all requests are made by an identified researcher user (and authorization decisions are based separately on that identity and local data entitlements).

## CanDIG federation governance model

The CanDIG platform's design has iteratively co-evolved with the governance of the CanDIG Federation, which defines the roles, responsibilities, and accountabilities of all stakeholders.

In our multi-institution, multi-project federated data effort, key stakeholders include the national platform, the data-hosting sites, and the data programs. All play a role in governance and setting the direction of the project. The national platform is responsible for technical decisions (on software development, architecture, standards, and data modeling) necessary for implementing a data platform consistent with requirements and convening discussions and building consensus around those requirements. The participating sites are responsible for maintaining the CanDIG software stack at their institution, connecting it to local infrastructure (such as compute and identity management), following operations and security policy and contributing to the development effort. The data programs are the custo-

dians for their local or distributed datasets, and own the relationships with the data subjects, are responsible for obtaining patient consents, data quality, and harmonization across their sites, working with the platform team to map to common data models, removal of direct identifiers, and communicating authorization decisions from their data access committees to the platform. Further details on roles and responsibilities are discussed in supplementary notes, section 3.

## CanDIG platform adoption of GA4GH standards and technologies

Making use of existing and emerging GA4GH standards and frameworks[3] where applicable helped CanDIG start quickly on its federation by giving us relevant technical, policy, and data model standards we could work with immediately. GA4GH standards were extremely current, and actively maintained, and so facilitated

collaborations such as our involvement in the Africa-Canada-EU CINECA project for federated analysis of human cohort data (web resources) which is committed to following GA4GH standards.

We prioritized standard adoption by how well they fit into our platform and federation design. As one example, an early task was to flesh out the roles, responsibilities, and accountabilities described above in greater detail. We used standards and frameworks, including the Framework for Responsible Data Sharing (web resources) and Data Security Infrastructure Policy (web resources) as a ready and comprehensive list of responsibilities for health data handling that we could ensure were clearly assigned to one stakeholder or explicitly shared between two stakeholders.

On the technical side, the first version of CanDIG APIs are built on top of the code for the initial Genomics APIs (GA4GH Server; web resources) developed at the University of California Santa Cruz for GA4GH. These Genomics APIs have now been discontinued by the GA4GH in favor of other standards, but using this code base allowed us to start working immediately with data custodians to make data available, and to build our authentication, authorization, and query federation framework atop of an existing codebase. Our authentication and authorization approach is described in supplementary notes section 1, including the adoption of an API gateway. Use of the gateway as a common external interface to our components allowed us to begin adding additional services and APIs, and replacing others, while making use of the same authentication, authorization, and query federation. This made it easier to adopt new services and APIs. Current core functionality of a CanDIGv1 site includes adopted GA4GH standards such as the Data Use Ontology (DUO),[6] which we use to document consents required for individual datasets; RNAGet (web resources) and htsget,[13] which we have implemented ourselves as standalone services, and Beacon,[10] as well as pre-existing standards that have been adopted by the GA4GH standards process such as CRAM/SAM/BAM, and VCF. In addition, DRS (web resources), and WES (web resources) are already being used internally at some sites.

The version currently in development, CanDIGv2 (web resources), includes those services as well as our implementations service-registry (web resources) to itemize the growing number of services available at a site, and Phenopackets (web resources) for structuring and returning phenotypic data for infectious disease or rare disease projects. In addition, Visa claims of the GA4GH Passport standard[5] are being used as a standard format to communicate data entitlements for a research user within a site. Finally, we are testing the use of GA4GH Variant Representation[4] as a common indexing mechanism for variants to solve the problem of allowing research users to perform variant queries in a number of different formats. Figure S1 shows how these tools and standards come together in the platform.

## CanDIG use by pan-Canadian projects

CanDIG currently makes genomic and phenotypic data available to scientists across Canada and international collaborators as part of data sharing for five leading pan-Canadian projects, including the Terry Fox Comprehensive Cancer Care Centre Consortium Network (TF4CN) and Terry Fox PRecision Oncology For Young peoPLE (PROFYLE[14]), as well as making human variant data from the Canadian COVID Genomics Network (CanCOGeN HostSeq; web resources) discoverable. It likewise makes data from provincial or single-site projects such as Personalized Onco-Genomics (Personalized Onco-Genomics; web resources) and the INSPIRE study[7] more accessible (current projects can be found listed on Table S1). These five projects, which include genomes and health data for nearly 2,000 study subjects, typically share their data via CanDIG so that users can discover subsets of relevant participant data ("data discovery") and explore that subset interactively.

CanDIG supports both controlled access and registered access research users. Controlled access is explicitly granted by data access committees, and researchers with controlled access entitlements can see and query (via the dashboard or programmatically via queries; see next session) significant amounts to those datasets. We also have a growing number of registered access users[15] who have signed up and agreed to terms of ser-

vice but have very limited querying ability, and only to those datasets (currently just CanCOGeN HostSeq) that have opted in to such access.

## Data access through CanDIG: Dashboard and queries

CanDIG provides web-based dashboards, and programmatic querying via APIs, of the datasets. Users generally start with the dashboard.

Initial panes of the dashboard include simple overviews of the data in a dataset such as count of data subjects by geography, demography, and broad phenotypic categories, as well as indicating what molecular data types (variants, reads, RNA expression) are available. These overviews are useful initial introductions to a dataset, and can be the main requirement for a project manager keeping track of the progress of a project; relatively modest levels of authorization are needed to be able to access the data for these panes (Figures 2A–2C).

Researchers with higher levels of controlled access can have deep access to the data, allowing them to dig into individual cases. This too can be done via the dashboard, which allows viewing mutations by gene (Figure 2D), integrated IGV for viewing variants and their sequencing context (Figure 2E), and information about the analysis pipeline producing those results (Figure 2F).

We also enable programmatic access to data in CanDIG via APIs. In addition to the APIs discussed above, CanDIG has implemented an initial set of cross-service queries that allow querying for patients that have given clinical, variant, and expression data features, integrating the results from multiple APIs. Use cases include programmatic data discovery—identifying and querying relevant subsets of data based on a set of criteria—in a potentially automated way, as well as data analytics. As an example, we have demonstrated the ability to use these APIs for privacy enhancing machine learning (Figure 1C and supplementary notes section 4). We have trained a classifier on genomic and clinical data that uses the cross-service counts query with our initial implementation of local differential privacy,[12] a method of privacy enhancing analytical queries with provable limits on leaking of private information based on perturbing query results.

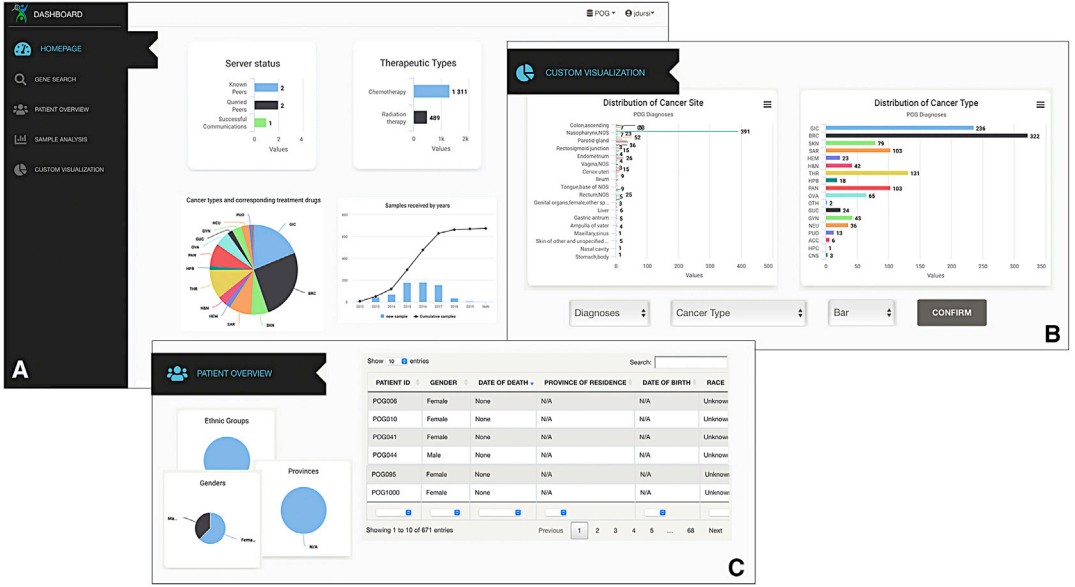

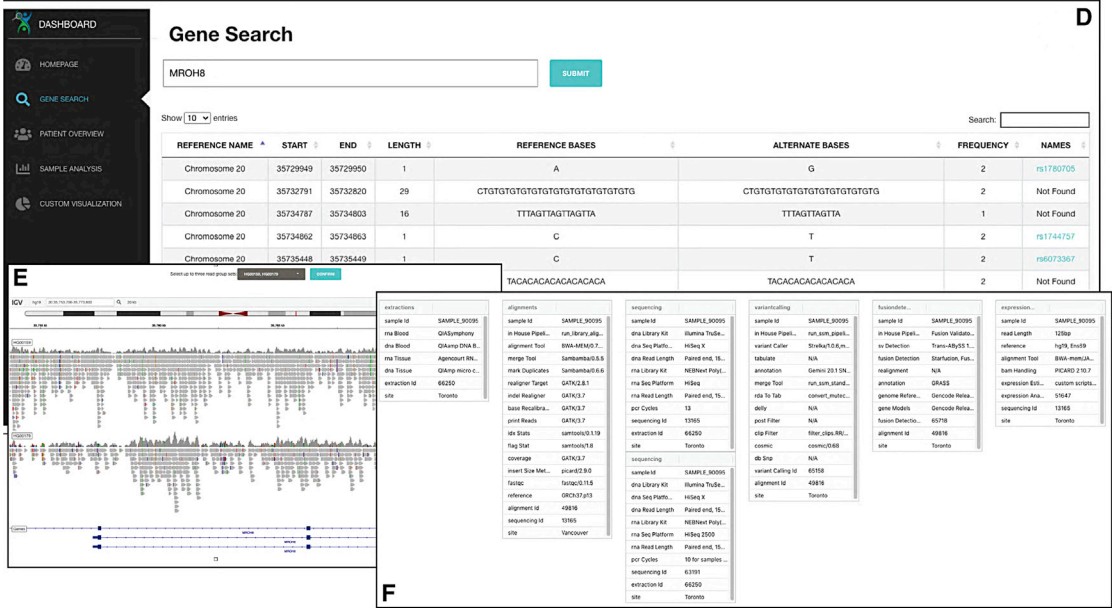

**Figure 2. Using the CanDIG portal**

CanDIG dashboard overview pages (for a research project manager, or someone examining a dataset for the first time), and more detailed pages (for authorized users). The website, and in particular the dashboard, is reactive and mobile-friendly, so that project status can be checked from anywhere.

(A–F) (A) The default simple summary statistics about participant data and cases available on the platform are shown. A project manager who wants to see breakdowns by other variables can use the custom visualization widget, illustrated in (B). Finally, the research project manager can quality check clinical data by a patient or check to see what associated genomic data is in the system through the patient's tab, illustrated in (C). A bioinformatician working with a particular sequencing project will want to explore genomic variants and read-level data, shown here with a federated mock dataset. A gene search page, shown in (D), allows querying variants in the dataset by gene name; from there the bioinformatician can view reads from the variant location in one or more participant's data in an embedded IGV.js tool, shown in (E). Information about the analysis pipeline used to generate the data from the sample collection to the variant calling tool settings can be shown through the sample analysis tab, shown in (F).

## Discussion

As the first Canadian national human genomics and biomedical federated data platform, CanDIG provides researchers, clinicians, and their international collaborators the ability to access and analyze a Canadian pool of multi-omics and health data and enables simpler data sharing across projects both nationally and internationally. By connecting multiple national health data projects in Canada on a single platform, and implementing GA4GH standards,

CanDIG enables all to benefit from pan-Canadian datasets, as CanDIG and international GA4GH-standards compliant efforts can be queried jointly. Our involvement with GA4GH and an international community involved in data federation developments, also allowed us to build these systems faster, taking advantage of lessons, development efforts, and components developed in other systems.

The governance model of our fully distributed, multi-jurisdiction, multi-project platform makes explicit the roles and responsibilities of the platform, software development effort, sites, and data custodians. The clarity and separation of roles greatly eases participation in international federation efforts such as with the EU/Canada/Africa CINECA project. We believe that our governance model is portable to a number of other distributed health data projects that share data among trusted partners.

Based on what we have learned, we are continuing to develop CanDIG with a more extensible service-oriented architecture, which will allow more ready incorporation of more services, such as further support for workflows via GA4GH WES (web resources), additional authorization capabilities, data access committee portals, additional molecular data types, and more complex analytics. We also look to enable interoperability with clinical data by moving to a standard clinical data model (OMOP: web resources), querying of medical imaging metadata, and stronger ontology support. These additional capabilities are necessary to support the upcoming Digital Health and Discovery Platform (DHDP: web resources) project, with greater volumes and variety of data—but the fundamental distributed authentication, authorization, and federation approach underlying CanDIG will remain unchanged.

### SUPPLEMENTAL INFORMATION

### WEB RESOURCES

Canada's Genomic Enterprise (CGen), https://www.cgen.ca/
GA4GH Framework for Responsible Sharing of Genomic and Health-Related Data, https://www.ga4gh.org/genomic-data-toolkit/regulatory-ethics-toolkit/framework-for-responsible-sharing-of-genomic-and-health-related-data/
GA4GH Data Security Infrastructure Policy, https://github.com/ga4gh/data-security/blob/master/DSIP/DSIP_v4.0.md
Common Infrastructure for National Cohorts in Europe, Canada, and Africa (CINECA), https://www.cineca-project.eu/
GA4GH Server, https://github.com/ga4gh/ga4gh-server
RNAget implementation, https://github.com/CanDIG/rnaget_service
GA4GH Data Repository Service (DRS) Schemas, https://ga4gh.github.io/data-repository-service-schemas/
GA4GH Workflow Execution Service (WES) Schemas, https://ga4gh.github.io/workflow-execution-service-schemas/
GA4GH Service Registry API, https://github.com/ga4gh-discovery/ga4gh-service-registry
Phenopackets, http://phenopackets.org/
CanDIGv2 software stack, https://github.com/candig/candigv2
CanCOGeN HostSeq, https://www.genomecanada.ca/en/cancogen/cancogen-hostseq
Personalized Oncogenomics Program (POG), https://www.bcgsc.ca/personalized-oncogenomics-program
GenAP, Genetics & Genomics Analysis Platform, https://genap.ca/
ID3 Decision Tree Classifier used to interact with the CanDIG server, implemented to allow differential privacy to protect PHI, https://github.com/CanDIG/id3-variants-training
OMOP-CDM, https://www.ohdsi.org/data-standardization/the-common-data-model/
DHDP, https://www.dhdp.ca

### ACKNOWLEDGMENTS

CanDIG development was funded by the Canada Foundation for Innovation Cyber infrastructure grant 34860, CANARIE Research Data Management contracts RDM-090 (CHORD) and RDM2-053 (ClinDIG), and the Canadian Institutes for Health Research as part of the Africa-Canada-EU Horizon2020 CINECA project (CIHR grant number #404896). P.E.J. is a research scholar from the Fonds de la recherche du Québec en santé (FRQS). G.B. is a Canada Research Chair in Computational Genomics and Medicine. M.B. is a CIFAR Canada AI Chair.

### AUTHOR CONTRIBUTIONS

Conceptualization, M.B., G.B., S.J.M.J., L.J.D., and C.V.; Software, L.J.D., Z.B., R.d.B., H.L., A.L., S.F.R., A.S., N.M., D.N., F.C.-S., M.W., Y.P., and A.P.; Resources, Z.L., P.-O.Q., M.B., C.V., S.J.M.J., and G.B.; Writing - Original Draft, L.J.D., Z.B., R.d.B., H.L., M.B., M.H., and S.P.; Writing - Review & Editing, L.J.D., S.P., M.B., M.H., Z.B., R.d.B., H.L., D.B., A.L., S.F.R., A.S., N.M., D.N., F.C.-S., M.W., P.-O.Q., Z.L., S.A., Y.P., A.P., M.H., K.P., S.A.G., S.H., L.L.S., D.M., C.V., T.J.P., P.-É.J., Y.J., S.J.M.J., and G.B.; Funding Acquisition, M.B., G.B., S.J.M.J., Y.J., P.-É.J., T.J.P., C.V., L.L.S., and D.M.; Project Administration: K.P. and S.P.; Visualization: S.P. and H.L.; Supervision: M.B., G.B., S.J.M.J., Y.J., P.-É.J., T.J.P., C.V., and L.J.D.

### DECLARATION OF INTERESTS

The authors declare no competing interests.

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
