## [Document S2. Transparent peer review records for Dursi et al. · Cell Genomics]

CanDIG: Federated network across Canada for multi-omic and health data discovery and analysis

L. Jonathan Dursi¹, Zoltan Bozoky², Richard de Borja³, Haoyuan Li⁴, David Bujold^{5,6}, Adam Lipski⁷, Shaikh Farhan Rashid¹, Amanjeev Sethi¹, Neelam Memon, Dashaylan Naidoo⁴, Felipe Coral-Sasso, Matthew Wong⁸, P-O Quirion^{5,6}, Zhibin Lu⁹, Samarth Agarwal¹⁰, Yuriy Pavlov¹, Andrew Ponomarev¹¹, Mia Husic¹², Krista Pace¹, Samantha Palmer¹, Stephanie A. Grover¹³, Sevan Hakgor¹⁴, Lillian L. Siu¹⁴, David Malkin¹⁵, Carl Virtanen⁹, Trevor J. Pugh^{16,14,3}, Pierre-Étienne Jacques¹⁷, Yann Joly¹⁸, Steven J. M. Jones^{4,20}, Guillaume Bourque^{19,21,22}, Michael Brudno^{23,1,24,*+}

Summary

Scientific Editor:	Orli Bahcall
Initial submission:	3/19/2021
Revision received:	8/13/2021
Accepted:	10/07/2021
Rounds of review:	2
Number of reviewers:	3

Referee reports, first round of review

Reviewer #1: This is an interesting article and software architecture for federated analysis of genomic data. I think it contributes towards showcasing reference implementations of federated architectures to help advance genomic data analysis to help address the challenge of performing such analysis in highly controlled environment. The article is fairly easy to read and understand for a technologist and incorporate ideas within other environments.

Minor Revisions:

1. Explain a little more about how the fine-grained AuthN/Z is implemented. There is a passing reference to GA4GH Passports & AAI, but the section that is relevant goes into describing RBAC and/or ABAC
2. Explain how the data is harmonised to allow such private queries to take place. This is an implicit assumption that needs to be teased out
3. It would be good to show which standards are being used in which version for CanDIG and which is not fully implemented. The paper makes WES + DRS + htsget implementation currently in development, so it would be appropriate to highlight it as such

Reviewer #2: The paper makes a good argument for the need for system to overcome the distributed nature of data and the difficulties of needing authenticated user access to controlled data. The paper makes this point too many times in the document, however. For example, even after writing the previous sentence, and

continuing to read the paper, the following sentence appears at the beginning of the discussion:

"Meanwhile, in distributed environments, especially those that are multi-jurisdictional, local and national project managers and data custodians need to ensure that their data remains securely and privately under the appropriate domain of control."

The authors claim that their system connects many institutions in Canada but does not offer any statistics about how widely it may be used -- e.g., how many process have been run on one of more of the local installations by researchers outside the local area? How many researchers have signed up to obtain credentials outside their locality? What specific kinds of analysis are being done on the system?

The system is Canada-specific, but might be useful more broadly as the issues address are not confined to Canada, e.g., varied jurisdictional criteria for data access. Will the code be configurable enough that it could be implemented elsewhere? There are very few technical details provided. How are credentials passed from one system to another?

"We believe that our approach is portable to a number of other distributed health data projects that share data among trusted partners."

It would be nice to hear more about the implementation and how it might be portable. The paper seems long on generalizations and short on implementation details.

=====

p.3

"1 We mean something quite specific here by data federation - querying and analyzing horizontal partitions of data, with participants geographically separated and all of their data located at one site."

something as important as the definition of "data federation" -- key to the paper -- should not be in a footnote. Also: "all of their data located at one site." is ambiguous. This reads as if the data are all at one location even though the participants are separated. Is this the intent? It would seem to define something more along the lines of centralized, not federated. Or do they mean, "all of their own data located at their own site" ?

--

"from having a few predefined queries, such as with the Beacon Network [9] and the Matchmaker Exchange (MME) [10], to running arbitrary workflows, such as with Datashield [11,12]. This, in"

Has the document swirthed from "author, year" reference format to [number]? If so , the refs do not have numbers at the end of the doc.

p.4

Figure 1. legend.

"C: A platform that is a decentralized, peer-to-peer network, where"

Is this also considered a federated platform? The subsequent sentence seems to imply such, but it is not stated. For A and B, "centralized" and "hub-and-spokes model of federation", respectively, it is named up front. Is it "A platform that is a decentralized, peer-to-peer federated network"?

--

"we'd refer to this as merely distributed rather than federated."

a useful distinction, but another important definition that might be better included in the text instead of a

figure legend.

p.7.

Table 1. It might be useful to explicate acronyms somewhere: e.g., CHUM, UHN, BCGSC

p.11

"CanDIGv1 was originally based on the University of California Genomics APIs"

which UC campus? there are several.

It is not clear from following the ref that it is a UC at all.

p.14

"However, the required access levels of each item - each property of any particular row - can be increased, allowing, for instance, data custodians responsible for data from particularly marginalized populations to require additional levels of authorization to access the study data."

why might it be necessary to increase restrictions on such data?

=====

nits

p.10

Beacon (Fiume et al., 2019), and htsgset (Kelleher et al., 2019), and shortly GA4GH Variant

probably should drop the "and" before "htsgset"

Fig3 legend

A schematic diagram illustrating how GA4GH technical standards and being used in the evolving CanDIG

"are" being used?

p.12

this sentence is awkward:

"To facilitate the wide range of research projects and queries we want to enable pushed us to support an increasingly rich range of datasets."

Reviewer #3: The CanDIG system is a technical prototype (ms, supplementary materials, p18) that contains less than 2,000 subjects in total from 4 studies (Table 1, ms p5). It is based upon a reference implementation, ga4gh-server, which has been retired (repository retirement notice, <https://github.com/ga4gh/ga4gh-server>).

CanDIG involves 5 institutions (ms p2), but currently only 3 sites (BCGSC, UHN, McGill) are actively participating (Table 1, ms p5).

As detailed in the notes below, the main ms doesn't provide enough technical details to describe the trade-offs and architectural choices made.

No actual use of the system is described.

Although a potentially interesting and important federated infrastructure, the work seems too preliminary for publication at this time.

Additional notes are below.

--

Page 2: "CanDIG is being used to make genomic and phenotypic data available for querying across Canada as part of data sharing for five leading pan-Canadian projects"

According to Table 1, only 3 sites are currently connected in the technical prototype.

Page 4: The approach proposed doesn't appear to be unique, so it may make sense to remove the word "unique" or to describe in what sense it is unique.

Page 4-5: Rather than describe the platform as national, it is probably more accurate to describe as "a project involving BCGSC, UHN, McGill, SickKids and CHUM." Or a project spanning "McGill University, The Hospital for Sick Children, University Health Network, Ontario Institute for Cancer Research, Canada's Michael Smith Genome Sciences Centre, Jewish General Hospital and Université de Sherbrooke."

Page 4-5 should simply describe what is set up now and what is its status (technical prototype, pilot, production, etc.) It would be helpful to list for each site what services they are running and what data they are hosting instead of talking in general terms about success. The current description is too general and not specific enough for a scientific paper.

Page 5 - the number of projects (4), the number of subjects per projects (570 subjects, 106 subjects, 20 subjects, 983 subjects) and the total number of subjects (< 2000) is very modest compared to other federated projects.

Page 7: "By adopting GA4GH standards, CanDIG allows for easy startup of national and international multi-site -omics projects where federated data queries and analysis can be made in a security-conscious environment. It makes genomics data available for querying and analyses through standard data models and modern ReSTful APIs. Making use of and participating in GA4GH standards development ensures that the solutions we build will be interoperable internationally, and allows us to take advantage of lessons, development efforts, and sometimes even entire components developed elsewhere. The current production CanDIG platform, CanDIGv1, implements GA4GH technical standards such as DUO6, RNAGet7, CRAM/SAM/BAM, and VCF,..."

This needs to be rewritten: Although CRAM/SAM/BAM and VCF have been adopted as GA4GH standards, these formats predate GA4GH and have been widely used and adopted and today are more of an artifact of the tools adopted than a decision to use GA4GH standards.

I'm not sure why participating in GA4GH standards discussions merits mentioning in this paper, nor mentioning standards that are not yet deployed such as DRS, WES, etc.

Page 8. "CanDIG's standards-based approach and use of modern tools implemented elsewhere extends to infrastructure authentication, where we use OAuth2 and OpenID Connect."

This is standard and not novel.

Page 9. "CanDIGv1 was originally based on the University of California Genomics APIs14, developed for an earlier version of the GA4GH effort, which allowed the CanDIG team to immediately make progress on federation."

I'm not sure this is relevant here, especially without mentioning specifically what technology. From the supplementary material, this is the ga4gh-server project, which has been retired.

Page 9-10. I don't find this discussion very precise or helpful. For example: "To facilitate the wide range of research projects and queries we want to enable pushed us to support an increasingly rich range of datasets. This, along with the range of data access that CanDIG enables, requires granular control over access to data and the development of services which can support multiple data types." Presumably the alternative is building a system that only supports one or two data types, but we are informed what the trade-offs if any for this decision vs alternatives.

Page 10. "In our decentralized federation, there is no central "CanDIG" identity; each user has a home site, typically the institution at which they work, where they can log in using their local credentials, and can view the dashboard there. This propagates the queries necessary to drive the dashboard across to all federation partners (Figure 1A). Currently a simple, single-step fan-out is sufficient. Each site in the federation recognizes the identity of the other sites and makes authorization decisions regarding access to the data it hosts. The granularity of that authorization allows for multiple projects to be supported without exposing data from one project to researchers of another unless so authorized. That information is then presented to the user via their home site. Because the user is necessarily authorized to see the data from the response, and the home site is effectively the user's work computer, the requests can be safely aggregated at the home sites. While this is our current approach, CanDIG is an ever-evolving platform, and other topologies have been tested and can be used as the number of our participating sites grows."

This sounds potentially interesting, but there is not enough detail to understand what is going on technically. Is some, all or no data centralized. What servers are being used? What APIs are used? Is this a technical prototype, a pilot?

Page 11. "The CanDIG platform provides authorized researchers access to complex queries and analyses of distributed datasets, jointly across various supported data types. This is enabled through very fine-grained authorization. Staff and researchers with different project roles or areas of specialty are granted different "access levels" to a dataset by the study's DAC, with each record of clinical and phenotypic data belonging to a dataset and each property within the record having a default access level. However, the required access levels of each item - each property of any particular row - can be increased, allowing, for instance, data custodians responsible for data from particularly marginalized populations to require additional levels of authorization to access the study data."

How is fine level authorization is done? What systems are used? What standards, if any? Is this centralized or distributed?

Page 12: "These components were unified through a single /search endpoint returning the information, or /count which returns counting aggregations."

What software is used? Is the same software used? What is the query language? How is the federation of results done?

Page 12. "For custodians who wish to make data available for discovery queries to a broader range of researchers while still maintaining participant privacy, CanDIG supports these queries as counts with local differential privacy (Duchi et al., 2013). Laplace or exponential noise is added to counts at each site before being summed and presented to the user (Figure 6A). This is the "local" addition of differentially private noise (as opposed to global differential privacy, Figure 6B), it allows the custodians of data at each site to determine the privacy-utility tradeoffs for the data under their control."

Was the software to implement differential privacy developed by CanDIG developed by CanDIG? If not, what software was used? Was the same software for differential privacy deployed at all sites? Is anyone actually using the differential privacy capability? If so, can that use be described? How as differential privacy explained to the IRB and where any problems encountered deploying differential privacy across the 5 institutions?

Page 13-14. If differential privacy used yet for any actual research? The use case described is over public data (1000 Genomes) and sounds more like a very simple proof of concept.

Page 14. "As the first Canadian national health genomics data federation platform, CanDIG lets researchers and clinicians in Canada access and benefit from a national pool of -omics data and enable much simpler sharing of data across projects - both nationally and internationally. By connecting multiple national health data projects in Canada on a single platform compliant with GA4GH standards, CanDIG is enabling all to benefit from the data being generated."

No actual use case was described of real use of the platform. The amount of data it seems available is quite small.

Page 14. "The governance of our fully distributed, multi-jurisdiction, and multi-project platform is complex. While the roles and responsibilities of different overlapping stakeholders can stay implicit in simpler centralized or single-dataset projects, in CanDIG they are made explicit. "

This is potentially interesting but not details are provided. OICR has been involved in quite complex international collaborations that require extensive legal agreements that seem more complex than what is described here, and what is novel here is not described.

Author response to the first round of review

12 Aug 2021

Dear Dr. Bahcall,

We thank the referees very much for their helpful comments - in particular their responses correctly pointed out that we tried to cover too much in the earlier manuscript, and as a result, there was a lack of clarity. Please find submitted here our resubmission, as a more focussed commentary giving an overview of CanDIG the federation effort, and how it was bootstrapped by taking advantage of the work and community of the GA4GH. We hope that the manuscript in its current form will be useful to other projects and efforts reading a special issue about the GA4GH and wondering how to leverage those efforts to start their own standards-based federation.

A point we tried to make in the previous version, and hopefully is clearer here (and well suited to the new focus) is that the federation roles and responsibilities, and governance, are really what define the federation. Software will be frequently rewritten as new capabilities are required - we have two manuscripts in preparation describing implementation details of CanDIGv2 - but the truly unique features of CanDIG are the definitions of the roles and responsibilities and the design of the structure of the federation.

Simultaneously, in this resubmission we have clarified technical discussion of the distributed nature of the authentication and authorization, and the service-oriented architecture of the stack, and added more text to the supplement. The service-oriented aspect is important and calling this out will hopefully clarify some confusion one referee had; unlike a centralized, monolithic platform where the team controlling the codebase also controls the place of deployment, data model, and data processing, our federation and service-oriented architecture allows (indeed, requires) significantly more flexibility in deployment, both across sites and of individual APIs. This service-based approach isn't novel, but is an important part of bootstrapping a national peer-led federation for genomics data in Canada.

In preparation for this resubmission we have

- Rewritten, shortened, reformatted, and streamlined the submission as a Commentary

- Expanded discussion of what is currently deployed, its status, and clarified what is not yet in production
- Added discussion of the technical implementation of the authentication and authorization, and how the service-based approach to deployment allowed iteration and heterogeneous deployments
- Ensured terms, concepts, and methods were clearly described on first use

Because of the extensive rewriting to reformat as a Commentary, we have not highlighted changed text in this resubmission; the changes were so extensive that most of the manuscript would have been highlighted.

Thank you, and all the best,
Jonathan Dursi and Michael Brudno

General Comments

1) The referees struggled in understanding how to review your manuscript, related in large part to the current presentation as well as insufficient technical details provided. This includes confusion over current status, and whether this is a technical prototype, pilot or platform, and at what stage are implementations. In additional comments to the editor, the referees noted that the current manuscript seemed more of a preliminary progress report than a scientific publication. Please revise robustly for both the content included and presentation for our broad readership as the reference publication of CanDIG.

We are hopeful the revised manuscript addresses these concerns.

2) Please format your manuscript according to our Commentary format.

We have shortened and streamlined the manuscript as a commentary. While we meet the word limit, we are over the citation count, as many point to the current special issue on GA4GH.

3) Please streamline the manuscript to focus on presenting the CanDIG platform and current implementations.

We are hopeful the revised manuscript addresses these concerns.

4) Please ensure that all technical and supporting information for CanDIG is included, as the reference publication for CanDIG. The referees were not able to fully review based on the information provided in the current manuscript.

We are hopeful the revised manuscript addresses these concerns. Some of the technical details are left to the supplement, for space reasons.

5) Please provide appropriate context for how CanDIG compares to other systems, with detailed comparisons including on technical implementations. This should be included (at least) in the introduction and discussion sections.

We have chosen to address this in Figures 1c-e and the corresponding text, as it connects to the functionality of CanDIG, as well as by putting some background text. We try to classify existing federation systems based on a number of dimensions. As specified in the discussion, and the response to reviewers, we are not aware of any federated systems that address the same problems CanDIG aims to solve.

6) Please provide clear description and status of the current as well as planned implementations. Please ensure that this is accurately reflected throughout the manuscript.

We are hopeful the revised manuscript addresses these concerns.

7) Please consider presentation for our broad readership. Please introduce terms, concepts and related methods/standards/work at first use.

We are hopeful the revised manuscript addresses these concerns.

Reviewer 1

This is an interesting article and software architecture for federated analysis of genomic data. I think it contributes towards showcasing reference implementations of federated architectures to help advance genomic data analysis to help address the challenge of performing such analysis in highly controlled environment. The article is fairly easy to read and understand for a technologist and incorporate ideas within other environments.

1. Explain a little more about how the fine-grained AuthN/Z is implemented. There is a passing reference to GA4GH Passports & AAI, but the section that is relevant goes into describing RBAC and/or ABAC

We've described this a little more in the body, particularly around authentication: e.g. in section "CanDIG's Use of GA4GH Standards and Technologies

By using existing tools developed and used by trusted organizations, CanDIG focused on its distinguishing requirements of a decentralized federation, including recognizing multiple identity providers at each site - a feature of Tyk, but implementable in other such gateways - and implementing authorization to consistently make use of distributed identities in local authorization decisions with local controlled access lists. This approach accelerated development and allowed us to leverage existing identity management solutions at each site. Some additional details are provided in the Supplement, section 1.

And in some more detail in the supplement in Section 1:

Key to our authentication are open-source software components: Keycloak and Tyk, one of which exists at each site. Managing the Keycloak instance is a shared responsibility between the CanDIG developers and the local host site IT team; it allows users to log in using research institution or hospital credentials, allowing trusted federation partners to handle identity management, while providing OpenID Connect identity tokens. Keycloak is connected to each site's internal authentication service and does not store any credentials itself. Keycloak has out of the box support for active directory, LDAP, and Kerberos for internal network user registries. We do not federate identities; we recognize as part of the federation agreement the distributed identities supplied by the federation partners. Relying on standard tools has made it easy to extend our approach to authentication - as part of the CINECA project, we have shown that we can conditionally accept ELIXIR OIDC tokens as an identity token. We had to fix a bug in Keycloak to do

it, but by doing so we contribute to the entire ecosystem of users interested in using GA4GH passport tokens.

Services rely on the user's OIDC Identity Token associated with each request - after a final validation - to authorize data requests. In the CanDIG model, authorization is informed by platform level information, but authorization decision making is always strictly local. Currently authorization information is maintained in a local text-based database; users with different project roles or areas of specialty are granted different "access levels" to a dataset by the study's DAC, with each record of clinical and phenotypic data belonging to a dataset and each property within the record having a default access level. However, the required access levels of each item - each property of any particular row - can be increased, allowing, for instance, data custodians responsible for data from marginalized populations to require additional levels of authorization to access the study data, reinforcing privacy protections for individuals particularly vulnerable to medical and other discrimination.

And text following. The approach proving solid, we are adding technical layers to the communication of platform-level authorization claims, so that we can integrate tools like the REMS DAC portal; for that we are using Passport "visas" to describe verifiably-signed claims.

2. Explain how the data is harmonised to allow such private queries to take place. This is an implicit assumption that needs to be teased out

We've added to body of the text, in the same section "CanDIG's Governance Model And Use of GA4GH Policies",

As an example of responsibilities and where they intersect, individual data projects and their custodians are responsible for within-project data quality and data harmonization, while the platform is responsible for defining overall data models that can support the range of projects, and such tools for translation as are feasible between the projects.

And detailed breakdown of the responsibilities is now in Supplementary Materials, Section 4.

3. It would be good to show which standards are being used in which version for CanDIG and which is not fully implemented. The paper makes WES + DRS + htsgat implementation currently in development, so it would be appropriate to highlight it as such

We've updated what is now Supplementary Figure 1, the diagram connecting the GA4GH standards, with what is a mandatory part of the federation described here (v1), services that are deployed at some sites but not yet a mandatory part of the deployment (v1+), and services that are part of the upcoming version two (v2) or are in development for early updates to that version (v2+).

We've also added some clarifying text into the body of the text (the same section) of what is being used now, even if not yet a mandatory part of the deployment, and what is upcoming.

Our decentralized federation does not require uniformity across sites - htsgat, RNAGet,

DRS, and WES are already being used internally at some sites. Testing of the upcoming next version, CanDIGv2, have shown that we can take immediately benefit from service-registry and Phenopackets, and that using OAuth2 and OpenID Connect technologies also allowed us to participate in and shape GA4GH standards development around the Passport (Voisin et al., 2021) which we are in the process of adapting for internal use. Supplementary Figure 1 shows how these tools and standards come together in the platform. A planned rewrite of a variant service will take advantage of the GA4GH Variant Representation (Wagner et al., 2021).

Reviewer 2

The paper makes a good argument for the need for system to overcome the distributed nature of data and the difficulties of needing authenticated user access to controlled data. The paper makes this point too many times in the document, however. For example, even after writing the previous sentence, and continuing to read the paper, the following sentence appears at the beginning of the discussion:

"Meanwhile, in distributed environments, especially those that are multi-jurisdictional, local and national project managers and data custodians need to ensure that their data remains securely and privately under the appropriate domain of control." The authors claim that their system connects many institutions in Canada but does not offer any statistics about how widely it may be used -- e.g., how many processes have been run on one of more of the local installations by researchers outside the local area? How many researchers have signed up to obtain credentials outside their locality? What specific kinds of analysis are being done on the system?

We've added the following clarifying text to the body of the paper, in the section "CanDIG Utilization in Pan-Canadian Projects"

CanDIG supports approximately 50 controlled access users and a growing number of registered access users accessing data from five projects, including genomes and health data for nearly 2,000 study subjects; users report that they currently use the platform for discovery of relevant data and subsequent interactive exploration.

The system is Canada-specific, but might be useful more broadly as the issues address are not confined to Canada, e.g., varied jurisdictional criteria for data access. Will the code be configurable enough that it could be implemented elsewhere? There are very few technical details provided. How are credentials passed from one system to another? "We believe that our approach is portable to a number of other distributed health data projects that share data among trusted partners." It would be nice to hear more about the implementation and how it might be portable. The paper seems long on generalizations and short on implementation details.

For the credentials, we've described this a little more in the body, particularly around authentication: e.g. in section "CanDIG's Use of GA4GH Standards and Technologies"

CanDIG focused on its distinguishing requirements of a decentralized federation, including recognizing multiple identity providers at each site - a feature of Tyk, but implementable in other such gateways - and implementing authorization to consistently make use of distributed identities in local authorization decisions with local controlled

access lists. This approach accelerated development and allowed us to leverage existing identity management solutions at each site. Some additional details are provided in the Supplement, section 1.

and we've added significantly more text into the referenced section of the supplement.

For the applicability to other federations or jurisdictions, the biggest question is whether the governance model is suitable. If so, our technical implementation is one instantiation of a design which may be applicable; if not, an entirely different approach would be required. In the section now moved to Supplement Section 4, we describe our model. In a situation where it is acceptable or desirable to devolve authorization decision making entirely to the local sites, the basic approach may well be suitable; indeed, in a forthcoming manuscript in preparation, we describe the authentication and authorization infrastructure for CanDIGv2, and will release it as a standalone product for those who may be interested in using it as a starting point. However, many federations may require control of authorization at a different level of the federation than the individual data-holding site; models that require central or regional control of authorization as part of the governance would find our technical implementation unsuitable.

We hope that the added description of methods in the Supplement are now suitable for the reworked manuscript, which is now a commentary with a focus on how GA4GH standards, tools, and policies helped us bootstrap a national cross-jurisdiction federation.

P.3 "1 We mean something quite specific here by data federation - querying and analyzing horizontal partitions of data, with participants geographically separated and all of their data located at one site." Something as important as the definition of "data federation" -- key to the paper -- should not be in a footnote. Also: "all of their data located at one site." is ambiguous. This reads as if the data are all at one location even though the participants are separated. Is this the intent? It would seem to define something more along the lines of centralized, not federated. Or do they mean, "all of their own data located at their own site"?

We thank the reviewer for their feedback, and have added this definition to the main body of the manuscript and amended the surrounding text to more clearly describe what we mean:

The word federation can cover a number of quite different arrangements (Thorogood et al., 2021). Here, we mean the connection of "horizontal partitions" of data - for instance, connecting geographically separated research cohorts where the data for various participants can be found at multiple sites. We do not, however, consider linking multiple separate data sources or types for the same data subject - clinical data in one store, genomic data in a second store, crossing "vertical partitions". In our model, this happens internally to a site, and we refer to those operations as performing data integration, rather than federation. We also distinguish between data that is merely distributed, falling upon a user to discover, query, and assemble results by themselves, and data within a federation, where the federation nodes coordinate and communicate amongst each other.

P.4 "C: A platform that is a decentralized, peer-to-peer network, where" Is this also considered a federated platform? The subsequent sentence seems to imply such, but it is not stated. For A and B, "centralized" and "hub-and-spokes model of federation", respectively, it is named up front. Is it "A platform that is a decentralized, peer-to-peer federated network"?

Thank you; we have amended the Figure 1 caption to better clarify our meaning:

A decentralized, peer-to-peer network, where a user sends a request to a peer health research centre - where relevant data may or may not be - and other peers are queried. Results (represented by DNA) are then returned to the user. We can further categorize well-known health data federations along dimensions of decentralization (hub-and-spokes to peer-to-peer).

P.4 "we'd refer to this as merely distributed rather than federated." A useful distinction, but another important definition that might be better included in the text instead of a figure legend.

We thank the referee for pointing these three items out and agree that the discussion is too important to be left to footnotes or scattered over different parts of the text.

In a revamped section entitled "Federation Models", we now aim to make this clearer:

As described in (Thorogood et al., 2021), the word federation can cover a number of quite different arrangements. Here, by federation we mean the connection of "horizontal partitions" of data - for instance, connecting geographically separated research cohorts where the different types of data for a given participant are all co-located, but the data for other participants can be found at other sites. We do not consider linking multiple separate data sources for the same data subject - clinical data in one store, genomic data in a second store, crossing "vertical partitions". In our model, this happens internally to a site, and we refer here to those operations as performing data integration rather than data federation. We also distinguish between data that is merely distributed, falling upon a user to discover, query, and assemble results by themselves, and data within a federation, where the federation nodes coordinate and communicate amongst each other.

"from having a few predefined queries, such as with the Beacon Network [9] and the Matchmaker Exchange (MME) [10], to running arbitrary workflows, such as with Datashield [11,12]. This, in" Has the document switched from "author, year" reference format to [number]? If so, the refs do not have numbers at the end of the doc.

Thank you for catching this. We have fixed the references in this resubmission.

P.7. Table 1. It might be useful to explicate acronyms somewhere: e.g., CHUM, UHN, BCGSC
In Table 1 (now in Supplement) these are now spelled out fully on first use.

P.11 "CanDIGv1 was originally based on the University of California Genomics APIs" Which UC campus? there are several. It is not clear from following the ref that it is a UC at all.

Thank you; in our rewrite, we have made sure to specify which campus.

The current CanDIG APIs make use of Genomics APIs developed at the University of California Santa Cruz for the GA4GH effort.

P.14 "However, the required access levels of each item - each property of any particular

row - can be increased, allowing, for instance, data custodians responsible for data from particularly marginalized populations to require additional levels of authorization to access the study data." Why might it be necessary to increase restrictions on such data?

Thank you; we have added text to clarify the importance of additional restrictions on such data, now in the supplement:

However, the required access levels of each item - each property of any particular row - can be increased, allowing, for instance, data custodians responsible for data from particularly marginalized populations to require additional levels of authorization to access the study data, reinforcing privacy protections for individuals particularly vulnerable to medical and other discrimination.

Beacon (Fiume et al., 2019), and htsget (Kelleher et al., 2019), and shortly GA4GH Variant probably should drop the "and" before "htsget"

Thank you; in our rewrite of this section, we have fixed this.

Fig3 legend A schematic diagram illustrating how GA4GH technical standards and being used in the evolving CanDIG "are" being used?

We thank the reviewer for catching this error and have implemented appropriate edits following the recommendation. As seen in Supplementary Figure 3 description:

A schematic diagram illustrating how GA4GH technical standards are being used in the evolving CanDIG platform as we move towards deploying CanDIGv2.

This sentence is awkward: "To facilitate the wide range of research projects and queries we want to enable pushed us to support an increasingly rich range of datasets."

Thank you; in our rewrite of this section, we have simplified and clarified the phrasing (underlined):

Improved understanding of health and disease requires contextualization of patient data through multi-omics approaches and research initiatives that integrate many data types. Supporting increasingly diverse datasets lets us facilitate a wide variety of research projects and queries. This, along with the range of data access that CanDIG enables, requires granular control over access to data and the gradual development of service-based architecture, allowing us to plug in additional services to support other data types while maintaining a single coherent policy and authorization framework at each peer site.

Reviewer 3

The CanDIG system is a technical prototype (ms, supplementary materials, p18) that contains less than 2,000 subjects in total from 4 studies (Table 1, ms p5). It is based upon a reference implementation, *ga4gh-server*, which has been retired (repository retirement notice, <https://github.com/ga4gh/ga4gh-server>).

CanDIG involves 5 institutions (ms p2), but currently only 3 sites (BCGSC, UHN, McGill) are actively participating (Table 1, ms p5).

That's correct. We have platform stakeholders - informing on governance, policy, contributing code, connecting us to other efforts (e.g. the GenAP platform) even at institutions that do not host sites.

As detailed in the notes below, the main ms doesn't provide enough technical details to describe the trade-offs and architectural choices made.

We hope the new focus of the revised commentary, on how the GA4GH standards, policies, and community helped us bootstrap a national federation, clarifies the choice of material we've included in this version of the manuscript.

No actual use of the system is described.

We've added the following clarifying text to the body of the paper, in the section "CanDIG Utilization in Pan-Canadian Projects"

CanDIG supports approximately 50 controlled access users and a growing number of registered access users accessing data from five projects, including genomes and health data for nearly 2,000 study subjects; users report that they currently use the platform for discovery of relevant data and subsequent interactive exploration.

Although a potentially interesting and important federated infrastructure, the work seems too preliminary for publication at this time.

We hope the new focus of the revised commentary, on how the GA4GH standards, policies, and community helped us bootstrap a national federation, makes for a more compelling manuscript.

--

P.2 "CanDIG is being used to make genomic and phenotypic data available for querying across Canada as part of data sharing for five leading pan-Canadian projects" According to Table 1, only 3 sites are currently connected in the technical prototype.

Yes, that's correct; five projects across three sites. In a federated data project, the number of projects need not, and in general will not have any relation to the number of sites.

P.4 The approach proposed doesn't appear to be unique, so it may make sense to remove the word "unique" or to describe in what sense it is unique.

We have removed the term following the reviewer's recommendation.

P.4-5 Rather than describe the platform as national, it is probably more accurate to describe as "a project involving BCGSC, UHN, McGill, SickKids and CHUM." Or a project spanning "McGill University, The Hospital for Sick Children, University Health Network, Ontario Institute for Cancer Research, Canada's Michael Smith Genome Sciences Centre, Jewish General Hospital and Université de Sherbrooke."

We're not sure we understand the distinction the referee is making here; we describe a nationally-funded (federal + match from three provinces) effort connecting the largest

sequencing centres in Canada - in particular the CGeN platform, which is certainly a national platform and involves a proper subset of the institutions involved in this work.

Should simply describe what is set up now and what is its status (technical prototype, pilot, production, etc.) It would be helpful to list for each site what services they are running and what data they are hosting instead of talking in general terms about success. The current description is too general and not specific enough for a scientific paper

We hope the new focus of the revised commentary, on how the GA4GH standards, policies, and community helped us bootstrap a national federation, clarifies the choice of material we've included in this version of the manuscript.

P.5 The number of projects (4), the number of subjects per project (570 subjects, 106 subjects, 20 subjects, 983 subjects) and the total number of subjects (< 2000) is very modest compared to other federated projects.

One of the most important benefits of interoperability powered by GA4GH standards - demonstrated by our participation in the CINECA project, or of any of a number of other international efforts - is that the connections and interoperability with other services and the data available leads to a network effect. This enables advances by making an ecosystem of data discoverable, queryable, and (by whatever mechanism) accessible to an analysis by researchers. Thus, while the actual number of cases in CanDIG is currently modest (albeit growing quickly with HostSeq and DHDP efforts), it can enable rapid scientific progress.

P.7 "By adopting GA4GH standards, CanDIG allows for easy startup of national and international multi-site -omics projects where federated data queries and analysis can be made in a security-conscious environment. It makes genomics data available for querying and analyses through standard data models and modern ReSTful APIs. Making use of and participating in GA4GH standards development ensures that the solutions we build will be interoperable internationally and allows us to take advantage of lessons, development efforts, and sometimes even entire components developed elsewhere. The current production CanDIG platform, CanDIGv1, implements GA4GH technical standards such as DUO6, RNAGet7, CRAM/SAM/BAM, and VCF,..." This needs to be rewritten: Although CRAM/SAM/BAM and VCF have been adopted as GA4GH standards, these formats predate GA4GH and have been widely used and adopted and today are more of an artifact of the tools adopted than a decision to use GA4GH standards.

We've updated the text to reflect this:

CanDIGv1 sites implement technical and data standards that originated with the GA4GH, such as the Data Use Ontology (DUO) (Lawson et al., 2021), RNAGet, htsget (Kelleher et al., 2019) and Beacon (Fiume et al., 2019), as well as others that are now part of the GA4GH standards process such as CRAM/SAM/BAM, and VCF.

I'm not sure why participating in GA4GH standards discussions merits mentioning in this paper, nor mentioning standards that are not yet deployed such as DRS, WES, etc.

We hope that the tighter focus of the resubmission, on how the GA4GH helped us bootstrap a genomic and health data federation, makes this inclusion clearer.

P.8 "CanDIG's standards-based approach and use of modern tools implemented elsewhere extends to infrastructure authentication, where we use OAuth2 and OpenID Connect." This is standard and not novel.

The referee is completely correct here. As suggested by the sentence's inclusion in a section on standards used by the project, and indeed by the quoted sentence itself in full ("*CanDIG's standards-based approach and use of modern tools implemented elsewhere extends to infrastructure authentication, where we use OAuth2 and OpenID Connect to ensure the privacy and security of all data, relying on well known implementations of Keycloak and Tyk*") we described relying on "well-known implementations" of OAuth2 and OpenID Connect: these are well known, trusted standards with mature implementations.

One of the keys to CanDIG's success has been consistently avoiding extraneous and costly novelty of implementation details except where absolutely necessary. Rather than reimplementing functionality existing elsewhere, we made an early decision to utilize and contribute to code of related efforts where appropriate, so we could focus on the genuinely hard problem of designing and deploying a first distributed genomics federation in context of the Canadian confederation.

In the current "CanDIG's Use of GA4GH Standards and Technologies" section, we emphasize this more clearly:

Making use of existing tooling and GA4GH standards helped CanDIG start quickly on its federation by enabling the team to quickly work with data custodians, and build trust with institutions and teams reluctant to make data more widely accessible.

The current CanDIG APIs make use of Genomics APIs developed at the University of California Santa Cruz for the GA4GH effort. While UCSC Genomics APIs have since been retired, bootstrapping from these allowed our team to immediately make progress on our bespoke federation components: implementing site-centric, peer-to-peer federation and governance, as well as distributed authentication and authorization. We intentionally avoided creating novel components except where necessary, using known standards such as OAuth2 and OpenID Connect, and relying on widely deployed implementations such as Keycloak and Tyk. By making an API gateway with single sign-on a part of our authentication and authorization infrastructure, we then could experiment with different services behind the gateway, and change implementations without affecting the interfaces. CanDIG implements technical and data standards that originated with the GA4GH, such as the Data Use Ontology (DUO) (Lawson et al., 2021), RNAGet, htsgget (Kelleher et al., 2019) and Beacon (Fiume et al., 2019), as well as others that are now part of the GA4GH standards process such as CRAM/SAM/BAM, and VCF.

Our decentralized federation does not require uniformity across sites - htsgget, RNAGet, DRS, and WES are already being used internally at some sites. Testing of the upcoming next version, CanDIGv2, have shown that we can take immediately benefit from service-registry and Phenopackets, and that using OAuth2 and OpenID Connect technologies also allowed us to participate in and shape GA4GH standards development around the Passport (Voisin et al., 2021) which we are in the process of adapting for internal use. Supplementary Figure 1 shows how these tools and standards come together in

the platform. A planned rewrite of a variant service will take advantage of the GA4GH Variant Representation (Wagner et al., 2021)

P.9 "CanDIGv1 was originally based on the University of California Genomics APIs, developed for an earlier version of the GA4GH effort, which allowed the CanDIG team to immediately make progress on federation." I'm not sure this is relevant here, especially without mentioning specifically what technology. From the supplementary material, this is the ga4gh-server project, which has been retired.

The rewrite shown above clarifies this; yes, the original codebase did come from the retired ga4gh-server project (and indeed pieces of that code base continues to be used).

P.9-10 I don't find this discussion very precise or helpful. For example: "To facilitate the wide range of research projects and queries we want to enable pushed us to support an increasingly rich range of datasets. This, along with the range of data access that CanDIG enables, requires granular control over access to data and the development of services which can support multiple data types." Presumably, the alternative is building a system that only supports one or two data types, but we haven't been informed what the trade-offs if any for this decision vs alternatives.

The quoted sentences have been removed in the course of the rewrite. We have aimed to make the discussion shorter, and more to the point during the edit process.

P. 10 "In our decentralized federation, there is no central "CanDIG" identity; each user has a home site, typically the institution at which they work, where they can log in using their local credentials and can view the dashboard there. This propagates the queries necessary to drive the dashboard across to all federation partners (Figure 1A). Currently, a simple, single-step fan-out is sufficient. Each site in the federation recognizes the identity of the other sites and makes authorization decisions regarding access to the data it hosts. The granularity of that authorization allows for multiple projects to be supported without exposing data from one project to researchers of another unless so authorized. That information is then presented to the user via their home site. Because the user is necessarily authorized to see the data from the response, and the home site is effectively the user's work computer, the requests can be safely aggregated at the home sites. While this is our current approach, CanDIG is an ever-evolving platform, and other topologies have been tested and can be used as the number of our participating sites grows." This sounds potentially interesting, but there is not enough detail to understand what is going on technically. Is some, all or no data centralized? What servers are being used? What APIs are used? Is this a technical prototype, a pilot?

There is no centralized data of any sort; we've made this more explicit in the "CanDIG federation model" section:

The Canadian Distributed Infrastructure for Genomics (CanDIG) provides a fully-distributed federated data platform for health research genomics data (and, increasingly, related data types) with no centralized infrastructure or data; coordination occurs through the collaboration of the sites and the policy and standards decisions at the platform level.

We describe in more technical details surrounding authentication and authorization in the body and supplement. What is described here is the production implementation of

the CanDIG federation. We apologize for lack of clarity in the earlier version of the manuscript.

P.11. "The CanDIG platform provides authorized researchers access to complex queries and analyses of distributed datasets, jointly across various supported data types. This is enabled through very fine-grained authorization. Staff and researchers with different project roles or areas of specialty are granted different "access levels" to a dataset by the study's DAC, with each record of clinical and phenotypic data belonging to a dataset and each property within the record having a default access level. However, the required access levels of each item - each property of any particular row - can be increased, allowing, for instance, data custodians responsible for data from particularly marginalized populations to require additional levels of authorization to access the study data." How is fine level authorization is done? What systems are used? What standards, if any? Is this centralized or distributed?

We've described this a little more in the body, particularly around authentication: e.g. in section "CanDIG's Use of GA4GH Standards and Technologies

By using existing tools developed and used by trusted organizations, CanDIG focused on its distinguishing requirements of a decentralized federation, including recognizing multiple identity providers at each site - a feature of Tyk, but implementable in other such gateways - and implementing authorization to consistently make use of distributed identities in local authorization decisions with local controlled access lists. This approach accelerated development and allowed us to leverage existing identity management solutions at each site. Some additional details are provided in the Supplement, section 1.

And in some more detail in the supplement in Section 1:

Key to our authentication are open-source software components: Keycloak and Tyk, one of which exists at each site. Managing the Keycloak instance is a shared responsibility between the CanDIG developers and the local host site IT team; it allows users to log in using research institution or hospital credentials, allowing trusted federation partners to handle identity management, while providing OpenID Connect identity tokens. Keycloak is connected to each site's internal authentication service and does not store any credentials itself. Keycloak has out of the box support for active directory, LDAP, and Kerberos for internal network user registries. We do not federate identities; we recognize as part of the federation agreement the distributed identities supplied by the federation partners. Relying on standard tools has made it easy to extend our approach to authentication - as part of the CINECA project, we have shown that we can conditionally accept ELIXIR OIDC tokens as an identity token. We had to fix a bug in Keycloak to do it, but by doing so we contribute to the entire ecosystem of users interested in using GA4GH passport tokens.

Services rely on the user's OIDC Identity Token associated with each request - after a final validation - to authorize data requests. In the CanDIG model, authorization is informed by platform level information, but authorization decision making is always strictly local. Currently authorization information is maintained in a local text-based database; users with different project roles or areas of specialty are granted different "access levels" to a dataset by the study's DAC, with each record of clinical and

phenotypic data belonging to a dataset and each property within the record having a default access level. However, the required access levels of each item - each property of any particular row - can be increased, allowing, for instance, data custodians responsible for data from marginalized populations to require additional levels of authorization to access the study data, reinforcing privacy protections for individuals particularly vulnerable to medical and other discrimination.

And text following. The approach proving solid, we are adding technical layers to the communication of platform-level authorization claims, so that we can integrate tools like the REMS DAC portal; for that we are using Passport “visas” to describe verifiably-signed claims.

P.12 "These components were unified through a single /search endpoint returning the information, or /count which returns counting aggregations." What software is used? Is the same software used? What is the query language? How is the federation of results done?

We've clarified that this is added to the candig-server code base, in what is now section 1 of the Supplement:

These components were unified through a single /search endpoint returning the information, or /count which returns counting aggregations, implemented in the candig-server code base.

The query language is an ad-hoc language informed by OData; now that the GA4GH Data Connect standard is formalized, we will look into that as a possible replacement.

P.12 "For custodians who wish to make data available for discovery queries to a broader range of researchers while still maintaining participant privacy, CanDIG supports these queries as counts with local differential privacy (Duchi et al., 2013). Laplace or exponential noise is added to counts at each site before being summed and presented to the user (Figure 6A). This is the "local" addition of differentially private noise (as opposed to global differential privacy, Figure 6B), it allows the custodians of data at each site to determine the privacy-utility tradeoffs for the data under their control." Was the software to implement differential privacy developed by CanDIG developed by CanDIG? If not, what software was used? Was the same software for differential privacy deployed at all sites? Is anyone actually using the differential privacy capability? If so, can that use be described? How was differential privacy explained to the IRB and where any problems encountered deploying differential privacy across the 5 institutions?

There has been interest in this demonstration - particularly the support for heterogeneous requirements across sites, which helps solidify the “local data-hosting sites control access to the data” model for data custodians. But as the referee suggests, DACs and IRBs are not yet familiar enough with privacy preserving methods to take advantage of this method, and our implementation (in the candig-server repo) remains available for further development for that reason; since initial development several open-source differential privacy libraries (e.g. <https://github.com/opensdp>) have been released which could be incorporated.

P. 13-14. If differential privacy used yet for any actual research? The use case described is over public data (1000 Genomes) and sounds more like a very simple proof of concept.

Indeed, this is a simple proof of concept. The differential privacy aspects are available to the data custodians to turn on if they wish; however it has not been used in practice as the policy aspects on how to set the relevant privacy parameters are not mature. Still, we believe it is important to demonstrate this technical functionality of the CanDIG platform, particularly the handling of heterogeneous privacy requirements, enabled by our federated approach.

We've clarified this in the text: *"A federated approach also allows the handling of sites with heterogeneous privacy requirements. For custodians who are considering make data available for discovery queries to a broader range of researchers while maintaining participant privacy, CanDIG has demonstrated support for these queries as counts with local differential privacy (Duchi et al., 2013) as a proof of concept."*

P.14. "As the first Canadian national health genomics data federation platform, CanDIG lets researchers and clinicians in Canada access and benefit from a national pool of -omics data and enable much simpler sharing of data across projects - both nationally and internationally. By connecting multiple national health data projects in Canada on a single platform compliant with GA4GH standards, CanDIG is enabling all to benefit from the data being generated." No actual use case was described of real use of the platform. The amount of data it seems available is quite small.

We discuss in the manuscript how the system can be used by a number of different "actors": e.g. study coordinators or bioinformaticians. Additionally, the focused rewrite of the manuscript as a commentary on how the GA4GH standards, policies, and community helped us bootstrap a national federation, hopefully clarifies the choice of material we've included in this version of the manuscript.

P.14. "The governance of our fully distributed, multi-jurisdiction, and multi-project platform is complex. While the roles and responsibilities of different overlapping stakeholders can stay implicit in simpler centralized or single-dataset projects, in CanDIG they are made explicit. "This is potentially interesting but no details are provided. OICR has been involved in quite complex international collaborations that require extensive legal agreements that seem more complex than what is described here, and what is novel here is not described.

We thank the referee for bringing up aspects of the platform that were not well described. We are hopeful that the new rewrite brings these novel aspects forward (though due to length constraints some details are left in the supplement.

While we cannot be 100% certain all of these are novel, any description at all of roles and responsibilities in a multi-custodian, multi-site health data federations are extremely hard to find in the literature. This gap greatly slows down progress - we can say with confidence that had that figure and table been handed to us on day one, we would be a year further ahead. Indeed, once the roles and responsibilities are clearly articulated and agreed upon, the most consequential technical decisions become straightforward.

Referee reports, second round of review

Reviewer #1: Thank you for article and it was very easy to peruse again. This is a very valuable and timely contribution to the development of a federated genomic analysis infrastructure. The article has provided examples of genemic analysis conducted in a federated environment and highlights some of the challenges especially in a highly controlled environment. I highly welcome this paper and happy with revisions addressed.

Reviewer #2: In general, the issues raised in the first review have been addressed. It still necessarily addresses an implementation of a long-recognized solution to the problem of data silos: federation of authentication. IT does not break much new ground, but is a useful description of a functioning, though limited-in-scope solution.

comments

.pdf pages

p1
"Code and Algorithms

Does your manuscript report custom computer code or introduce a new algorithm?" Yes.

I'm not sure it does.

p25
"technical and data standards ... as well as others that are now part of the GA4GH standards process such as CRAM/SAM/BAM, and VCF."

might be ore accurately phrased:

"technical and data standards ... as well as pre-existing standards that have been adopted by the GA4GH standards process such as CRAM/SAM/BAM, and VCF."

needs to be more clear that they did not originate w GA4GH

small nits - these are small, but the number of them reflect an apparent low level of close scrutiny. This undermines confidence in the rest of it. And they are distracting.

.pdf pages

p20
"CanDIG was built by leveraging the standards and tooling and frameworks brought together by the Global Alliance for Genomics and Health (GA4GH), allowing the project to implementing international standards"

should be:

(GA4GH), allowing the project to implement international standards

p20

"CanDIG was built by leveraging the standards and tooling and frameworks brought together by the Global Alliance for Genomics and Health (GA4GH), allowing the project to implementing international standards while being able to focus on the particular needs of its decentralized federation and defining a clear division of responsibilities among participants, which includes participation from McGill University, The Hospital for Sick Children, University Health Network, Ontario Institute for Cancer Research, Canada's Michael Smith Genome Sciences Centre, Jewish General Hospital and Université de Sherbrooke."

all one sentence?

p25

"Testing of the upcoming next version, CanDIGv2, have shown..."

has shown...

"Testing of the upcoming next version, CanDIGv2, have shown that we can take immediately benefit"

awkward. rephrase

p28

"For custodians who are considering make data available for discovery queries "

"For custodians who are considering making data available for discovery queries "

p29

"based on 17 known ancestry informative SNPs on five the 1000 genomes superancestries,"

"based on 17 known ancestry informative SNPs on five of the 1000 genomes superancestries,"

p34

"Key to our authentication are open-source software components"

"Key to our authentication IS open-source software components"

p34

"out of the box"

"out-of-the-box"

Reviewer #3: Comments enter in this field will be shared with the author; your identity will remain anonymous.

I think the paper is much improved, but what has been achieved seems relatively standard and the implementation seems still to be very much a prototype at this time.

Author response to the second round of review

18 Oct 2021

Dear Dr Bahcall:

Please find attached a revised version of manuscript CELL-GENOMICS-D-21-00082R1, with almost-total rewrites of all sections aimed at improving the clarity and focus of the paper, following your and the referee's extensive suggestions. Because of significant changes in this response, we provide both a marked up and a "clean" version of the manuscript, in Word and Google Docs format, and the updated supplement. Our next step is to go through the Final Files Checklist.

Our abstract is now 60 words, the paper is down to 15 references and 2 display items, we have a web resources section, and the body of the text (Introduction to Acknowledgements inclusive, omitting figure captions) is now 2,911 words. We use terms consistent with those used in the Commentary on Federation, and refer repeatedly to the GA4GH papers in the special issue.

Our responses to the referees follow:

Reviewer #1: Thank you for the article and it was very easy to peruse again. This is a very valuable and timely contribution to the development of a federated genomic analysis infrastructure. The article has provided examples of genomic analysis conducted in a federated environment and highlights some of the challenges especially in a highly controlled environment. I highly welcome this paper and am happy with revisions addressed.

Thank you for your comments.

Reviewer #2: In general, the issues raised in the first review have been addressed. It still necessarily addresses an implementation of a long-recognized solution to the problem of data silos: federation of authentication. It does not break much new ground, but is a useful description of a functioning, though limited-in-scope solution.

Thank you for your comments.

p1

"Code and Algorithms Does your manuscript report custom computer code or introduce a new algorithm?" Yes. I'm not sure it does.

Thanks for your comment. We definitely report custom computer code, but agree that there are no new algorithms.

p25

"technical and data standards ... as well as others that are now part of the GA4GH standards process such as CRAM/SAM/BAM, and VCF."

might be more accurately phrased:

"technical and data standards ... as well as pre-existing standards that have been adopted by the GA4GH standards process such as CRAM/SAM/BAM, and VCF."

needs to be more clear that they did not originate w GA4GH

Thanks: we have made exactly that change in "CanDIG Platform Adoption of GA4GH Standards and Technologies". The current sentence now reads "Current core functionality of a CanDIGv1 site includes adopted GA4GH standards such as the Data Use Ontology (DUO)[6], which we use to document consents required for individual datasets; RNAGet (web resources) and htsgat[11] which we have implemented ourselves as standalone services, and Beacon[9], as well as pre-existing standards that have been adopted by the GA4GH standards process such as CRAM/SAM/BAM, and VCF"

small nits - these are small, but the number of them reflect an apparent low level of close scrutiny. This undermines confidence in the rest of it. And they are distracting.

p20

"CanDIG was built by leveraging the standards and tooling and frameworks brought together by the Global Alliance for Genomics and Health (GA4GH), allowing the project to implementing international standards"

should be:

(GA4GH), allowing the project to implement international standards

Thanks for pointing this out; in the rewritten, shorter, abstract we have removed this sentence entirely.

p20

"CanDIG was built by leveraging the standards and tooling and frameworks brought together by the Global Alliance for Genomics and Health (GA4GH), allowing the project to implementing international standards while being able to focus on the particular needs of its decentralized federation and defining a clear division of responsibilities among participants, which includes participation from McGill University, The Hospital for Sick Children, University Health Network,

Ontario Institute for Cancer Research, Canada's Michael Smith Genome Sciences Centre, Jewish General Hospital and Université de Sherbrooke."

all one sentence?

Thanks for your comment; the first part of this sentence has been deleted; the second part, now in the introduction, reads "CanDIG Participants include McGill University, The Hospital for Sick Children, University Health Network, Ontario Institute for Cancer Research, Canada's Michael Smith Genome Sciences Centre, Jewish General Hospital and Université de Sherbrooke."

p25

"Testing of the upcoming next version, CanDIGv2, have shown..." has shown...

"Testing of the upcoming next version, CanDIGv2, have shown that we can take immediately benefit" awkward. Rephrase

Thanks for pointing this out; in the now rewritten and shorter section now titled "CanDIG Platform Adoption of GA4GH Standards and Technologies," the offending sentence the two comments above refer to has been removed.

p28 "For custodians who are considering make data available for discovery queries "

"For custodians who are considering making data available for discovery queries "

In the rewritten, shorter, and merged section now titled "Data Access through CanDIG - Dashboard and Queries", this sentence has been removed.

p29 "based on 17 known ancestry informative SNPs on five the 1000 genomes superancestries,"

"based on 17 known ancestry informative SNPs on five of the 1000 genomes superancestries,"

Thank you for correcting this; the figure caption sentence, now in the Supplementary Materials, now correctly reads "demonstrates a proof of concept, training of an ID3 tree based on 17 known ancestry informative SNPs on five of the 1000 genomes" as you've suggested.

p34

"Key to our authentication are open-source software components"

"Key to our authentication IS open-source software components"

Thanks; this text has been removed, and discussion of key open-source software components for authentication is now confined to the Supplementary Materials.

"out of the box" "out-of-the-box"

Thank you - this sentence in the supplement now reads "Keycloak has out-of-the-box support for active directory..."

Reviewer #3:

I think the paper is much improved, but what has been achieved seems relatively standard and the implementation seems still to be very much a prototype at this time.

Thanks for your comments; we're pleased that you find the paper improved, and your earlier comments have greatly strengthened the communication of this work.